

# Home hazard modification programs for reducing falls in older adults: a systematic review and meta-analysis

Charupa Lektip[1,2], Sirawee Chaovalit[3], Apichai Wattanapisit[4,5],
Sarawut Lapmanee[6], Jiraphat Nawarat[1,2] and Weeranan Yaemrattanakul[3]

[1] Movement Sciences and Exercise Research Center, Walailak University,
Nakhon Si Thammarat, Thailand
[2] Department of Physical Therapy, School of Allied Health Sciences, Walailak University, Nakhon Si Thammarat, Thailand
[3] Department of Physical Therapy, Faculty of Medicine, Prince of Songkla University, Songkhla, Thailand
[4] School of Medicine, Walailak University, Nakhon Si Thammarat, Thailand
[5] Walailak University Hospital, Nakhon Si Thammarat, Thailand
[6] Department of Basic Medical Sciences, Faculty of Medicine, Siam University, Bangkok, Thailand

## ABSTRACT

**Objective**. This study aims to assess the effect of home modification in preventing falls in older adults.

**Methods**. A systematic review and meta-analysis of randomized studies were performed. The review was conducted according to the Preferred Reporting Items for Systematic Reviews and Meta-Analyses (PRISMA) guidelines and was registered prospectively. Five electronic databases were systematically searched for related articles. The titles and abstracts of the articles found using the key search phrases—home modification and falling—were screened using inclusion and exclusion criteria. The Cochrane risk of bias tool was used to evaluate the studies' methodology.

**Results**. A total of 12 trials were included. A meta-analysis was conducted using 10 studies with $n = 1,960$ participants showing a clinically meaningful 7% reduction in falls (risk ratio = 0.93; 0.87–1).

**Conclusions**. Falls can be significantly reduced with the use of home modification interventions that are thorough, well-focused, have an environmental-fit perspective, and have adequate follow-up.

# INTRODUCTION

Falls are a significant cause of injuries and death in older adults, leading to ∼36,000 deaths and $50 billion for yearly medical costs (*Centers for Disease Control and Prevention, 2023*). The older adults who fall can suffer terrible consequences. It might lead to persistent discomfort, and loss of independence. The problem of falls among older adults has grown as a result of the aging population and is projected to become epidemic in scope (*Vaishya & Vaish, 2020*). Moreover, older adults may develop a fear of falling, resulting in activity restriction and a decline in physical and cognitive functions, which may affect the quality

Corresponding author
Sirawee Chaovalit,
csirawee@medicine.psu.ac.th

of life (*Schoene et al., 2019*). Multifactorial therapies, such as exercise prescription, physical activity promotion, education, and home modification, are used by healthcare practitioners to reduce falls and the risk of falls among the older adult (*Hopewell et al., 2020*).

Environmental factors are the major cause of falls, particularly falls occurring at homes (*Zhang et al., 2019*; *Sophonratanapokin, Sawangdee & Soonthorndhada, 2012*). Home modification is a potential intervention to reduce fall risks in this population (*Hill et al., 2018*). The term "home modification" is used to describe the act of changing settings to reduce the number of accidents and promote independent living (*McCullagh, 2006*). Previous systematic reviews show evidence for both single- and multi-component treatments that included home modifications to reduce the number and risk of falls among older adults and to improve function for people with a variety of health conditions. (*Stark et al., 2017*). *Cumming et al. (1999)* designed randomized study found that among older adults living in the community who were at high risk for falling, a home hazard reduction program did not reduce their risk of falls. The intervention was successful in reducing the rate of falls as a secondary outcome. A specified secondary outcome of the intervention, a lower rate of falls, was achieved. Additionally, another systematic review reveals the positive outcomes of home modification interventions in enhancing participation for community-dwelling individuals and older adults (*Chase et al., 2012*). A meta-analysis of randomized trials on environmental interventions to prevent falls in community-dwelling older people showed a 21% decrease in the incidence of falls (relative risk (RR) = 0.79; 0.65 to 0.97). The significant treatment impact of one trial was the cause of the heterogeneity. An analysis by *Clemson et al. (2008)* found a clinically significant 39% reduction in falls (RR = 0.61; 0.47 to 0.79), representing an absolute risk difference of 26% for a number needed to treat four individuals (*Clemson et al., 2008*).

Although evidence suggest that home modification can enhance the performance and safety of older adults with functional impairments. Nevertheless, previous evidence showed some ambiguous results. A meta-analysis of current scientific evidence would be beneficial for fulfilling this knowledge gap of the effectiveness of home modification interventions. Thus, the goal of this systematic review and meta-analysis was to update prior reviews, investigate the evidence, and characterize the effectiveness of the evidence for the usefulness of environmental interventions in preventing falls, improving the precision of known results.

### Survey methodology

The preferred Reporting Items for Systematic Reviews and Meta-Analyses (PRISMA) guidelines (*Moher et al., 2009*; *Liberati et al., 2009*), registered with the International Prospective Register of Systematic Reviews (PROSPERO; http://www.crd.york.ac.uk/prospero/, reference CRD42021286049) was used to perform the analysis.

### Search strategy

Search term for the population of interest were "elderly people" OR "older adult*". Search terms for the type of intervention were "home safety" OR "home modification" OR "home hazard". Search terms for the outcome parameter were "fall*" OR "risk of falls"

**Table 1  Inclusion and exclusion criteria.**

| Categories | Inclusion | Exclusion |
|---|---|---|
| Populations | ● Elderly people (aged 60 and over) who are healthy or with multiple disabilities, such as vision impairment, hearing loss, neurological or musculoskeletal disease, or cognitiveimpairment.<br>● Elderly living in residential settings. | ● Elderly people living in home health care or places other than the residential settings. |
| Intervention | ● Environmental interventions include removal of home hazards, home safety modifications to decrease falls and improve safety in activities of daily living. | ● Multicomponent interventions that included home modifications less than 75% |
| Outcomes | ● The number of fallers or rate of falls. | ● Effect on other people (e.g., parent, caregiver, teacher) |
| Type of studies | ● Randomized controlled trials<br>● Full text | ● Case studies<br>● Qualitative studies<br>● Article with abstracts only<br>● Thesis |

OR "accidental falls". The search was performed using the MEDLINE, PEDro, CINAHL, Scopus, and OTseeker databases after PROSPERO registration. It was repeated to update for new research results on January 2023. Results were limited to randomized controlled trial based on the Cochrane Highly Sensitive Search Strategy (*Lefebvre, Manheimer & Glanville, 2011*) using filters. The full search strategy is given in Table S1.

## Study selection

Inclusion and exclusion criteria for study selection are given in Table 1. Two reviewers (CL and SC) independently screened titles and abstracts of all search results to select studies of interest, *i.e.,* clinical trials on home modification programs in older adults. If the data included in the abstract wasn't clear two reviewers (CL and SC) separately read the full texts of these studies. Cohen's Kappa ($\kappa$) with 95% confidence level was used to measure the degree of agreement between reviewers. A score of $\kappa > 0.6$ was seen to be a substantial level of agreement (*Orwin, 1994*). Any disagreement was resolved by discussion with a third reviewer (SL). The full article was written in English, non-randomized clinical trials and articles not related to the aging population were excluded as well as articles that did not apply a home modification program to an extent of more than 75% (assume from the type of providing 100% fall prevention program, the program must be home modification less than 75%) or that did not report on falling as the primary outcome.

## Data extraction

Items of data extraction and risk of bias correspond to the previous systematic review (*Chaovalit, Taylor & Dodd, 2020*), which evaluated the effect of interventions from randomized controlled trials to increase research knowledge. A data extraction form according to the template for intervention description and replication checklist (TIDieR) (*Hoffmann et al., 2014*) was used to summarize all information of interest: study objective, study design, subject details and the description of the intervention in the experimental and control groups as well as full reference details. One reviewer (CL) extracted the data. A second reviewer (SC) checked for accuracy.

## Risk of bias assessment

The risk of bias with respect to selection, performance, detection, attrition and reporting in the studies included was assessed using the Cochrane risk of bias tool (*Higgins et al., 2011*). Completeness of the description of the intervention in the experimental and control groups was assessed using the TIDieR checklist (*Hoffmann et al., 2014*). The review was performed independently by two reviewers (CL and SC). The level of agreement was calculated and measured with Cohen's $\kappa$ levels. Any disagreement was resolved by discussion with a third reviewer.

The Grading of Recommendations Assessment, Development, and Evaluation (GRADE) criteria were applied to assess the quality of all meta-analyses (*Atkins et al., 2004*; *Puhan et al., 2014*). If one or more studies included in an analysis did not provide allocation concealment or blinded outcome assessment, in case of a moderate level of statistical heterogeneity between the trials ($I^2 > 50\%$), or if confidence intervals exceeded 0.8 standardized mean differences, quality was downgraded from high to moderate. If two of these three criteria were met, quality was downgraded from high to low.

## Data analysis

For dichotomous outcomes, intervention effects for each study were calculated using risk ratios, and 95% confidence intervals (CI) were shown in a forest plot. When at least three trials had comparable populations, interventions and controls, and outcome measures and were clinically homogeneous, a meta-analysis was carried out using Review Manager (RevMan) version 5.3 (*The Cochrane collaboration, 2008*). With risk ratio results, meta-analyses were done using a random-effects model. The results favored the group receiving the home modification intervention, as demonstrated by the risk ratio <1.

## RESULTS

### Identifying eligible trials

Of the 2,108 identified articles (including duplicates), 31 articles were retrieved for the full-text review after the title and abstract screening. After full-text review, 20 articles were excluded because two were not randomized controlled trials, 8 showed the extent of the home modification program as <75%, 7 were not reported falling as the primary outcome, one article was not completely provided in the English language, and two were not related to aging population; 11 studies met the inclusion criteria. The updated search identified five additional articles; of these, one article met the inclusion criteria, and four articles were excluded because one was not a randomized controlled trial, two showed the extent of the home modification program as <75%, and one was not reported falling as primary outcome (Table S2—the detail of the articles excluded after full-text review). In total, the results of 12 trials were included in the final review (Fig. 1).

### Risk of bias

Overall quality varied from low to moderate (Table 2; Fig. 2). Most studies had a minimal risk of bias due to appropriate sequence creation, incomplete outcome data, and selective outcome reporting. Five of 12 trials indicated a significant risk of allocation concealment

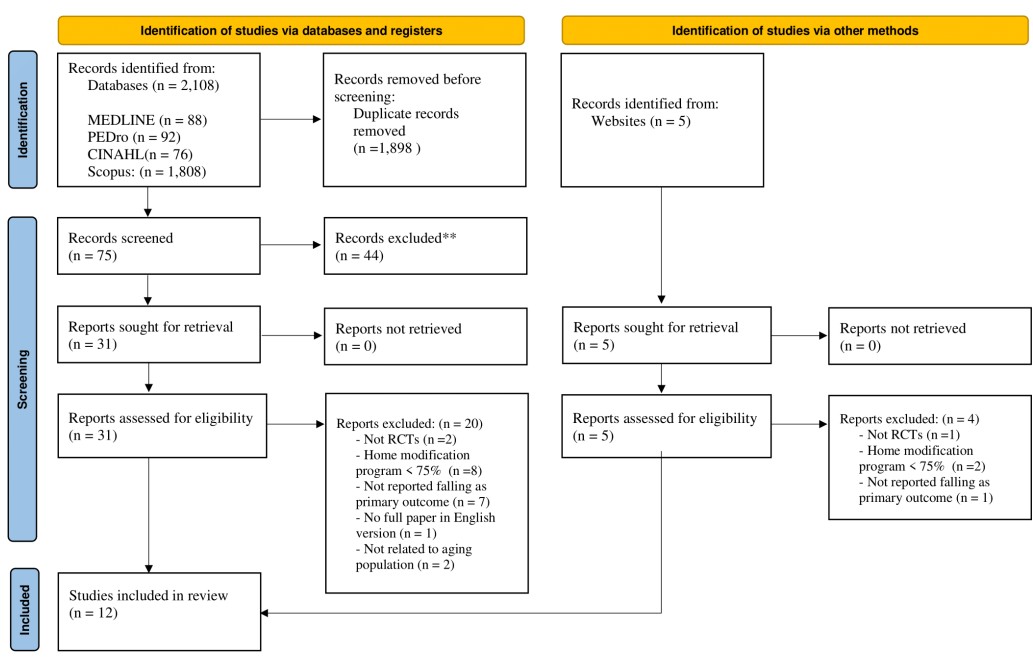

**Figure 1** Flow diagram of study selection.

**Table 2** Risk of bias summary.

| Authors | Adequate sequence generation | Allocation concealment | Blinding participant and personal | Blinding outcome assessment | Incomplete outcome data | Selective outcome report | Other source of bias |
|---|---|---|---|---|---|---|---|
| *Stark et al. (2021)* | L | L | L | U | L | L | L |
| *Cockayne et al. (2021)* | L | L | H | H | L | L | L |
| *Chu et al. (2017)* | L | L | L | L | L | U | H |
| *Kamei et al. (2015)* | L | H | H | H | L | L | H |
| *Pighills et al. (2011)* | L | H | H | L | L | L | H |
| *La Grow et al. (2006)* | U | L | U | H | H | H | H |
| *Campbell et al. (2005)* | U | L | U | H | H | H | H |
| *Nikolaus & Bach (2003)* | L | L | H | H | L | L | L |
| *Pardessus et al. (2002)* | U | H | H | H | U | U | L |
| *Stevens et al. (2001)* | L | L | H | L | L | L | H |
| *Gerson, Camargo Jr & Wilber (2005)* | H | H | H | H | L | U | L |
| *Cumming et al. (1999)* | L | U | H | L | U | L | H |

bias, indicating the possibility of selection bias. Nine of 12 indicated a high risk of ascertainment bias from participant and personal blinding, whereas seven of 12 reported blinding outcome assessment, indicating the possibility of ascertainment bias. There was broad agreement among researchers ($\kappa = 0.81$, 95% CI [0.68–0.94]).

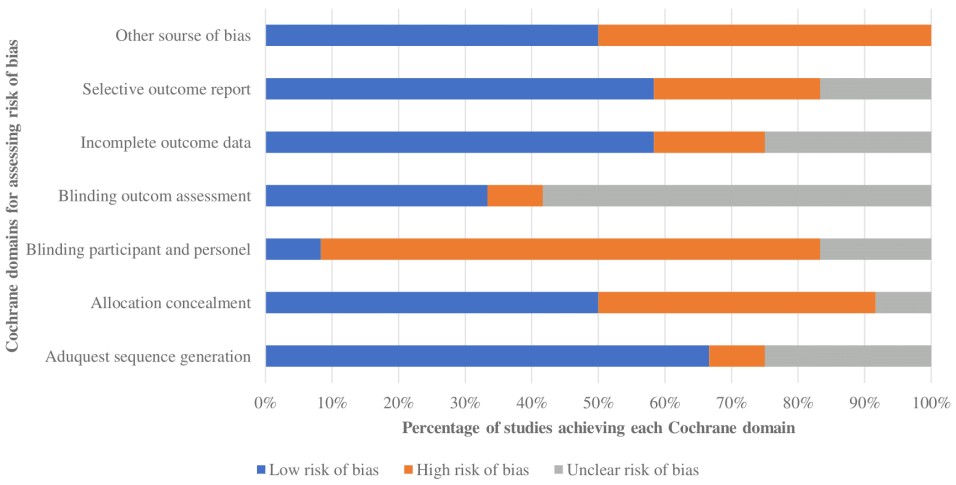

**Figure 2   Risk of bias of trials included in the systematic review.**

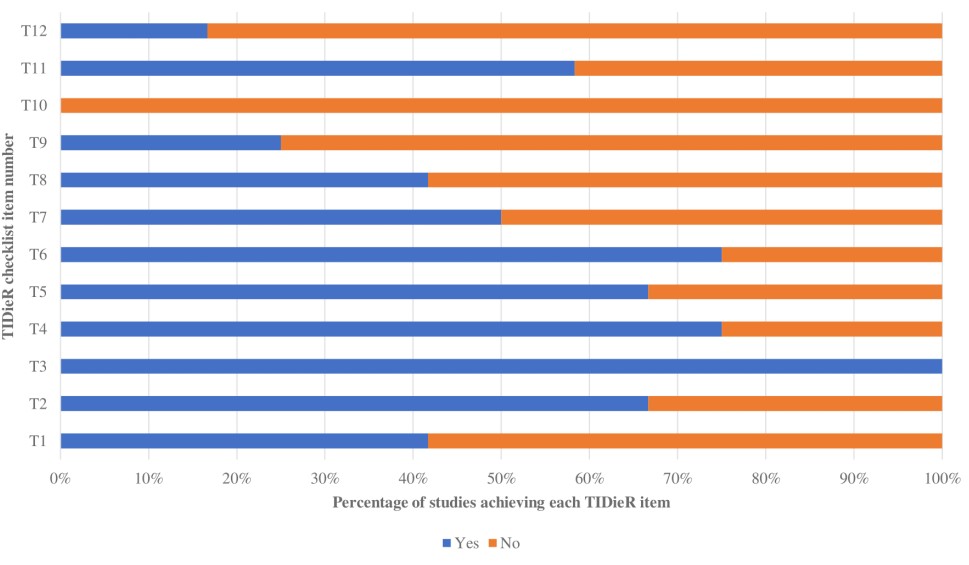

**Figure 3   Percentage of included studies achieving each TIDieR item of the experimental group.**

## Completeness of reporting

Using the TIDieR checklist, the intervention group's shown method (TIDieR item 3, T3) was best described in the presented information. Two of the least favorably reported elements were the modification of the intervention during the study (T10) and evaluation of intervention adherence (Table S3—the TIDieR table). The intervention group trials were reported more thoroughly than those of the control group, as shown in Figs. 3 and 4. For the intervention group, the percentage of trials qualifying for each TIDieR item ranged from 0% to 100%, whereas for the control group, the percentage ranged from 0%

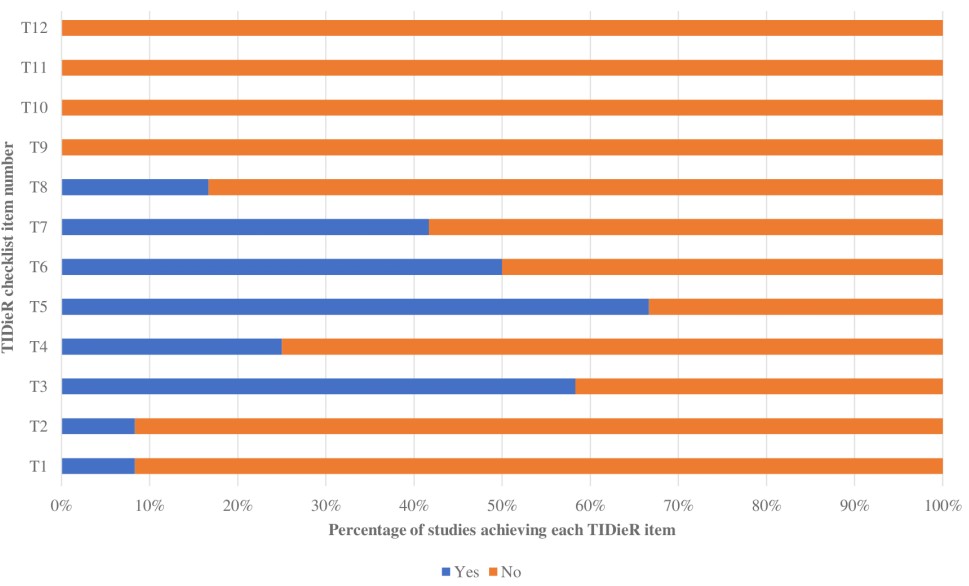

**Figure 4  Percentage of included studies achieving each TIDieR item of the control group.**

to 66.67%. There was broad agreement among researchers ($\kappa = 0.83$, 95% CI [0.74–0.92]).

## Trial characteristics

The characteristics of the study are listed in Table 3. The average age of most of the participants was between 70 and 79 years, with only three studies including participants in the late elder population (≥80 years) (*Campbell et al., 2005*; *Nikolaus & Bach, 2003*; *Pardessus et al., 2002*). Seven studies with follow-up periods ranging from 1 to 12 months evaluated adherence to an intervention (*Stark et al., 2021*; *Chu et al., 2017*; *Pighills et al., 2011*; *Campbell et al., 2005*; *Gerson, Camargo Jr & Wilber, 2005*; *Nikolaus & Bach, 2003*; *Pardessus et al., 2002*). The remaining five studies did not involve intervention adherence (*Cockayne et al., 2021*; *Kamei et al., 2015*; *La Grow et al., 2006*; *Stevens et al., 2001*; *Cumming et al., 1999*). Most interventions begin with an assessment of home hazards using reliable assessment methods, followed by occupational therapist-provided home improvement suggestions. Two studies offered various interventions. *Kamei et al. (2015)*, using a 60 cm × 60 cm residential model that the public health nurse researcher provides lectures and uses the mock-up to teach about a home hazard awareness program and education on how to adjust and create safety in a residential environment. Subjects participated in interactive practice with the instructor by removing obstacles from the mock-up to ensure the safety of the floor and environment in each location. *Gerson, Camargo Jr & Wilber (2005)* presented brochure recommendations that included physical activity, vision screening, medication assessment, and house modification.

## Effect of home modification on falls

After applying meta-analysis to 10 trials (1,960 participants), we obtained moderate-quality evidence that home modification was effective for reducing the number of falls
**Table 3  Characteristics of included studies[*].**

| Study | Group (I/C) | Sample size, Sex | Age (Y), (Mean ± SD) | Program detail | Outcome Measures | Results |
|---|---|---|---|---|---|---|
| Stark et al. (2021) | I | 155; 122 F, 33 M | 74.7 ± 7.4 | Three sessions provide to intervention group consist of 1) home hazard assessment using The Westmead Home Safety Assessment. (2) Facilitated home modifications home modification by interventionist (3) Complete installation and training. | • Falls • Daily activity performance | • No difference for outcome of fall hazard. • Thirty-eight percent reduction in the rate of falling in the intervention group compared with the control group. • No difference in daily activity performance. |
| | C | 155; 122 F, 33 M | 75.1 ± 7.7 | Usual care | | |
| Cockayne et al. (2021) | I | 430; 285 F, 185 M | 79.9 ± 6.4 | The Westmead Home Safety Assessment was used to examine home hazards, and an OT modify home relate danger. | | |
| | C | 901; 587 F, 314 M | 80.2 ± 6.3 | Usual care from their GP and other health-care professionals | | |
| Chu et al. (2017) | I | 95; 60 F, 30 M | 78.6 ± 6 | The OT Fall Reduction Home Visit Program consisted of an environmental hazard's evaluation and the Westmead Home Safety Assessment to identify environmental hazards, and a daily life routine assessment. The follow-up telephone call regarding home modification and assistive devices 2 months after the home visit. | • Number of fallers and repeated fallers • Number of falls and recurrent falls • Time until first fall | • The percentage of fallers over 1 year was 13.7% in the IG and 20.4% in the CG. • Significant differences in the number of fallers ($\rho$ = 0.03) and the number of falls ($\rho$ = 0.02) between the two groups over 6 months. • Significant differences in survival analysis for first fall at 6 months, but not 9 or 12 months. |
| | C | 103; 76 F, 27 M | 78.1 ± 6.1 | A single visit by a research assistant who had no professional training and no knowledge of fall prevention | | |
| Kamei et al. (2015) | I | 67; 56 F, 11 M | 75.7 ± 6.7 | HHMP group was provide for intervention group consist of a residential safety self-assessment for assessing home hazard and using a displayed 60 cm × 60 cm residential mock-up for practice. | • Falls • Fall prevention awareness. • Home modification | • The HHMP group achieved a 10.9% reduction in overall falls compared with the control group. • Significant increase in fall prevention awareness in the HHMP group between baseline and 52 weeks ($\rho$ <0.05). • The highest rates of modification were addressing "clutter" on the floor (82.1% in HHMP and 61.1% in control) |
| | C | 63; 9 F, 63 M | 75.8 ± 6.4 | Knowledge about falls risk, safety, and nutrition. A short talk on health and aging including demonstrate foot care, were provide by physician research and nurse. | | |

Lektip et al. (2023), *PeerJ*, DOI 10.7717/peerj.15699

**Table 3** (*continued*)

| Study | Group (I/C) | Sample size, Sex | Age (Y), (Mean ± SD) | Program detail | Outcome Measures | Results |
|---|---|---|---|---|---|---|
| Pighills et al. (2011) | I | 87; 62 F, 25 M | 78 ± 5 | Home hazard assessment, using the Westmead Home Safety Assessment, conducted in the home. Potential fall hazards were discussed, The OT suggested possible solutions and agreed on recommendations. A follow-up telephone contact was made after 4 weeks, and another telephone contact was made after 12 months. | | |
| | C | 78; 52 F, 26 M | 80 ± 7 | Usual care | • Falls<br>• Quality of life<br>• Barthel Index | • Fall rate in the OT group was approximately half that in the control group.<br>• No difference was found between OT groups and the control group in independence in the QOL ($\rho = 0.98$). |
| La Grow et al. (2006) | I | 100 | NR | A home safety checklist, using modified version of the Westmead home safety assessment checklist, was provided by an OT. The OT facilitated provision | Falls (hazard and non-hazard related) | Hazard-related and non–hazard-related falls were reduced in the home safety group compared with the control group. |
| | C | 96 | NR | | | |
| Campbell et al. (2005) | I | 100; 66 F, 24 M | 83.1 ± 4.5 | Home safety program included a home safety checklist using a modified version of the Westmead home safety assessment checklist with referral and recommendations to reduce home hazards and adherence using a telephone interview six months after study entry by an OT. | | |
| | C | 96; 67 F, 29 M | 84 ± 4.9 | Sixty-min social visits in the home during the first six months of the trial. | • Fall and injury<br>• Program cost | • Home safety program participants had 41% fewer falls than those who did not receive this program.<br>• No significant difference was found in reduction in falls at home compared with falls away from the home.<br>• The home safety program cost $432 per fall prevented. |
| Gerson, Camargo Jr & Wilber (2005) | I | 118; 67 F, 51 M | 75.4 ± 6.8 | Two brochures produced by the Centers for Disease Control and Prevention consist of (1) checklist of home hazards (2) the suggestions included exercise, vision check, medication review, and home modification. During the 1-month telephone call they would be asked questions about the brochures. | | |
| | C | 279; 180 F, 99 M | 75.7 ± 7.1 | A telephone call and be asked questions about home safety and receive information about how they could make their home safer. | • Falls<br>• Self-report of change in the home | • Almost half of ED population reported having had a fall in the prior year.<br>• 11 percent of patients who fell made a change in home compared with 6% of those who did not fall. • 9% of the control group and 8% of the intervention group made safety changes to the home environment. |
| Nikolaus & Bach (2003) | I | 181; 131 F, 50 M | 81.2 ± 6.3 | Home visit to inform about the possible fall risks in their home, to give advice on possible changes of the home environment, to facilitate any necessary home modifications, and to teach the persons in the use of technical and mobility aids when necessary. One year later, home visits were made for all participants. | | |
| | C | 133; 131 F, 2 M | 81.0 ± 6.5 | Not receive any type of home visit | • Number of falls<br>• Type of recommended home modifications<br>• Compliance with recommendations | • 163 falls in the intervention group and 204 falls in the control group.<br>• 31% lower fall rate in the intervention group than for the control group, but the proportion of frequent fallers did not significantly differ between the groups. |
| Pardessus et al. (2002) | I | 30; 23 F, 7 M | 83.51 ± 9.08 | Home visit to identify environmental hazards, and modifications were recommended. Simple home hazard removal was accomplished during assessment, if possible. Whenever a hazard could not be removed, the OT provided advice on how to live more safely with the hazards. Follow-up was provided by phone every month for 6 month and at 12 months. | | |
| | C | 30; 24 F, 6 M | 82.9 ± 6.3 | Therapeutic modifications from PT during hospitalization and those were informed on home safety and possible social assistance. | • Falls<br>• Fall-related institutionalization and death at 6 and 12 months. | • 28 patients had a fall recurrence; 15 were in the control group and 13 in the intervention group but not significant in the different.<br>• The mean number of fall recurrence was 0.72 ± 0.19 (control group, 0.82 ± 0.22; intervention group, 0.68 ± 0.16; not statistically significant).<br>• The rate of falls, institutionalization, and death were not significantly different between the two groups. |

Lektip et al. (2023), *PeerJ*, DOI 10.7717/peerj.15699

**Table 3** (*continued*)

| Study | Group (I/C) | Sample size, Sex | Age (Y), (Mean ± SD) | Program detail | Outcome Measures | Results |
|---|---|---|---|---|---|---|
| *Stevens et al. (2001)* | I | 570; 306 F, 264 M | 76 NR | The intervention consisted of three strategies: a home hazard assessment, the installation of free safety devices, and an educational strategy to empower elderly to remove or modify home hazards. | | • No significant reduction in the intervention group in the incidence rate of falls. |
| | C | 1,167; 602 F, 565 M | 76 NR | Acted to reduce the number of fall hazards in homes, having been alerted by the daily calendar to the purpose of the study and potential causes of falls. | Falls | • No reduction in the rate of all falls or the rate of falls inside the home. • No significant reduction in the rate of injurious falls in intervention subjects. |
| *Cumming et al. (1999)* | I | 264; 149 F, 115 M | 76.4 ± 7.1 | Home visit and gave specific recommended home modifications conducted by an OT. Program adherence by the OT telephoned about 2 weeks after visit. | | • 36% of subjects in the intervention group had at least one fall during follow-up, compared with 45% of controls ($\rho$ = .05). |
| | C | 266; 154 F, 112 M | 77.2 ± 7.4 | No home visits. | Falls | • The intervention was effective only among subjects ($n = 206$) who reported having had one or more falls during the year before recruitment into the study. |

**Notes.**

I, intervention group; C, control group; NR, not reported; M, male; F, female.

*The review was performed after PROSPERO registration and repeated to ensure the update new research on January 2023.

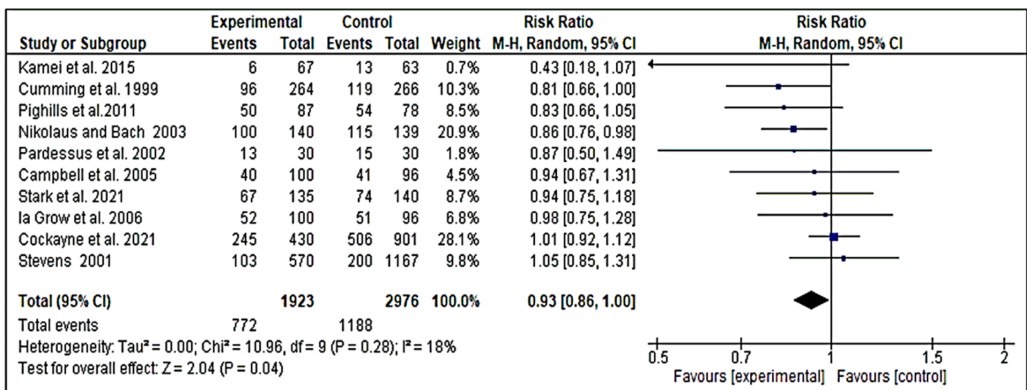

**Figure 5** Forest plot: effect of home modification on falling by pooling data from ten studies.

compared with the usual care (*Stark et al., 2021*; *Chu et al., 2017*; *Kamei et al., 2015*; *Pighills et al., 2011*; *La Grow et al., 2006*; *Campbell et al., 2005*; *Gerson, Camargo Jr & Wilber, 2005*; *Nikolaus & Bach, 2003*; *Pardessus et al., 2002*; *Cumming et al., 1999*). Two other trials (*Cockayne et al., 2021*; *Stevens et al., 2001*) could not be included in this meta-analysis. Both trials measured home modification in the emergency department (ED) population. The risk ratio of the 10 trials was 0.93 [0.86, 1.00, $I^2 = 18\%$] (Fig. 5).

## DISCUSSION

This systematic review and meta-analysis showed the effects of home environmental intervention in preventing falls. Our findings indicated a 7% reduction in the risk of falls across all studies. Only two of 10 studies (*Cockayne et al., 2021*; *Nikolaus & Bach, 2003*) found that house modifications did not decrease falls. This may be due to the lack of intervention adherence in both studies. The findings of these studies have implications for policy, practices, and resource allocation, and a pragmatic recommendation is made that the older adults at risk be provided with environmental condition is diagnosed by experienced specialists. One or two visits to the house may be sufficient to lower the likelihood of a fall-prevention intervention, which may be cost-effective and advantageous. According to *Campbell et al. (2005)*, the cost of the home safety program per fall averted was $432 (*Campbell et al., 2005*).

This study enhances prior literature reviews by giving current data for home modification methods for preventing falls in older adults. In accordance with prior studies, this study indicated that home modifications reduce falls among older persons at high risk (*Chase et al., 2012*; *Clemson et al., 2008*). Because of various methods for home modification, this study grouped all studies by the target population to evaluate the efficacy of home modifications. Additionally, this investigation was limited to studies that evaluated the efficacy of interventions for falling as the primary outcome. This study provides professionals with the most recent evidence-based practice in the field of environmental modifications considering specific outcomes and populations. Moreover, to enable standardized and comparable outcomes, guidelines must be set for home modification

intervention study. These guidelines should include follow-up period, environment, and outcome measurements, reporting of results, and operational definitions of home modification treatments.

We developed a guidance for health professionals, older adults, and caregivers for what constitutes a high-quality home modification intervention for preventing falls in older adults. Despite its seeming convenience, identifying specific change-related risks represents just the beginning of the complexity. Hence, a full examination of home hazard tools is required. Numerous studies (*Romli et al., 2018*) have developed a home hazard assessment to reduce falls in the older adult. Each evaluation tool's validity and reliability are reported so that it can be effectively implemented. The Westmead Home Safety Assessment was implemented by the majority of the studies in this investigation to investigate home hazards. This instrument's content validity index was 0.80, and inter-rater reliability was fair to good (0.48–1.00). The assessment contained a checklist of 13 home hazards including external trafficways, general indoors, mobility aids, internal trafficways, living area, seating, bedroom, footwear, pets, bathroom, toilet area, kitchen, medication management, safety call system (*Clemson, Fitzgerald & Heard, 1999a*; *Clemson et al., 1999b*). The interaction between an individual and their environment consists of risk-taking behaviors, protective behaviors, functional vision, and physical and cognitive characteristics that influence falling. This study demonstrates the importance of interventions that manage these variables. Seldom have studies explicitly focused on these variables for preventing falls. Besides that, almost all studies' findings support that good adherence produces positive outcomes to prevent falls.

Environmental interventions should be incorporated into predischarge planning for high-risk patients and postdischarge follow-up for those who have a history of falls, according to this study. We propose that older patients treated in emergency rooms following a fall should also be included in this high-risk group. Evidence supports a complete strategy for fallers who present to the emergency room, including home evaluations and occupational therapy intervention (*Cockayne et al., 2021*; *Stevens et al., 2001*).

Currently, there is evidence that home modification treatments that are comprehensive, well-targeted, and have an environmental-fit perspective can help reduce falls among the older adult high-risk groups. It seems likely that a compensatory approach consisting of raising awareness, enhancing safety measures, and reducing hazards in the home would be useful for older individuals who are at risk. Assessing the elders and their caregivers, making recommendations, and providing sufficient training, education, and follow-up for successful home modification interventions to promote involvement are the responsibilities of health professionals.

A possible limitation of this review is that the meta-analysis on home modification included a wide range of achievement measures, which could provide a problem for the homogeneity of the outcome measures used in this research. Nonetheless, we believed that falling was a primary outcome of home modification. Moreover, the $I^2$ score of 18% indicated that the statistical heterogeneity in this meta-analysis was minimal. Specific outcomes examined in the many individual trials, such as daily activity performance,

measured by specific measures, such as the Barthel Index, could not be applied to meta-analyses. The systematic review presented here has several strengths. This review's ability to precisely adhere to the PRISMA standards is one of its strongest points. Additionally, the GRADE technique was used to assess the degree of evidence of the results between trials, as presented in a meta-analysis. All trials were evaluated for reporting completeness using the TIDieR checklist. Further, the methodology included a large time frame (since 1999) and covered several bibliographic databases, thereby guaranteeing that relevant literature was captured.

## CONCLUSIONS

There is moderate-quality evidence from 10 randomized controlled trials that applied home modification to prevent falling in older adults. Across all studies showed a 7% reduction in the risk of falls. The home modification intervention should be:

- standardization
- well focused both person and environmental fit
- adequate follow up
- sufficient training and education for the older adults and their caregivers
- add the intervention to be part of the normal discharge procedure after referral from emergency department.

### Funding
The authors received no funding for this work.

### Competing Interests
The authors declare there are no competing interests.

### Author Contributions

- Charupa Lektip conceived and designed the experiments, performed the experiments, analyzed the data, prepared figures and/or tables, authored or reviewed drafts of the article, and approved the final draft.
- Sirawee Chaovalit conceived and designed the experiments, performed the experiments, analyzed the data, prepared figures and/or tables, authored or reviewed drafts of the article, and approved the final draft.
- Apichai Wattanapisit conceived and designed the experiments, analyzed the data, prepared figures and/or tables, authored or reviewed drafts of the article, and approved the final draft.
- Sarawut Lapmanee conceived and designed the experiments, analyzed the data, prepared figures and/or tables, authored or reviewed drafts of the article, and approved the final draft.
- Jiraphat Nawarat analyzed the data, authored or reviewed drafts of the article, and approved the final draft.

- Weeranan Yaemrattanakul analyzed the data, authored or reviewed drafts of the article, and approved the final draft.

## Data Availability

This study is a systematic review, so there are no raw data or code.

## Supplemental Information

Supplemental information for this article can be found online at http://dx.doi.org/10.7717/peerj.15699#supplemental-information.

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
