# Peer review of "Home hazard modification programs for reducing falls in older adults: a systematic review and meta-analysis"

_PeerJ, doi:10.7717/peerj.15699_

## Round 0.1 · original submission · Minor Revisions

Please take particular note of referencing errors and incorrect values noted by Reviewers 2 and 3.

Reviewer 1 ·

Basic reporting

Thank you for allowing me to review your manuscript. The topic of preventing falls in older adults through home modification is important and timely. Although clear and professional English is used throughout, there are a few areas that could be improved and I will list my recommendations here:
1. Avoid the use of the term "elderly". It is considered to be ageist or a term in which a stereotype is promulgated. [Avers et al (2011) Journal of Geriatric Physical Therapy] . Older adult or older person is a respectful term.
2. Line 41 - leave out "In 2020" as this situation occurred in years other than 202o as well. The sentence could state Falls are a significant..."
3. Line 53 - replace "the significant results of evidence that identified" with "show evidence for..."
4. Line 56-59 is not clear and should be rewritten.
5. Line 64 should read "An analysis by Clemson and colleagues (2008) found a ..."
6. Line 67 should read "Although evidence suggest..., results are ambiguous and there remain questions."
7. Line 73 - end the sentence after falls.
8. Lines 252-259 are not clear and should be rewritten for clarity.

The literature references are sufficient to provide background on falls in older adults. However, some are dated. There is abundant literature on falls in older adults and I believe you could update some of these references.

The article structure, figures, and tables are appropriate.
Make sure the search diagram confirms to the PRISMA 2020 statement. (http://prisma-statement.org/prismastatement/flowdiagram.aspx?AspxAutoDetectCookieSupport=1)
Your data is shared through the summary table of articles. It would be helpful for you to include a justification for the dates chosen for your review.

Your results and conclusions are relevant to your research question.

Experimental design

The research question is well defined and relevant. The search was performed to a high technical and ethical standard with sufficient detail to replicate.

There are some areas that need further detailed description.
1. Line 93 - define "extent <75%".
2. Line 177 - Define the 60 cm x60 cm residential model.
3. I recommend adding more details about the Westmead Home Safety Assessment used.

Validity of the findings

The conclusions are well stated and linked to original research question.
Line 194 - I recommend leaving this paragraph out as it does not pertain to your research questions or supported by results. I recommend the same for lines 223-230.

Additional comments

The discussion section and conclusions could be more concise. perhaps it would be helpful to include some bullet points of your recommendations such as
standardization
follow up
referral from emergency department

Reviewer 2 ·

Basic reporting

Thank you very much for the opportunity to review this manuscript. I have reviewed “Home hazard modification program for reducing falls in older adults: a systematic review and meta-analysis” with interest.
This systematic review can be highly evaluated as one of the multifactorial intervention programs for the fall prevention for the older adults, and as a manuscript that updates the evidence of interventions to maintain safety at home for older adults with a high risk of falling. The review complies with PRISMA, and I acknowledge that it is a very clear and easy-to-understand paper. Discussion part is also considered valid.

Experimental design

No comment.

Validity of the findings

No comment.

Additional comments

The following is a minor comment, do consider these corrections.
1. Please use the family name for citations in both the text and meta-analyses. Some are quoted by First name and mixed with First name and Family mane citations. It is necessary to correct to unify with the Family name.
2. In the reference list, please mark the references used for meta-analysis with an asterisk.

Reviewer 3 ·

Basic reporting

The findings of this paper are significant because environmental factors contribute to many falls experienced by older adults living at home. After reviewing the paper, I found numerical inaccuracies and incorrectly cited references. Unfortunately, I cannot accept it due to these technical inaccuracies.

Experimental design

No comment

Validity of the findings

No comment

Additional comments

Examples of numerical inaccuracies and incorrectly cited references are the following:

Reference
× 34)Tomoko K, Fumiko K, Yuko Y, Yukako I, Rumi K, Tomoko S, et al. Effectiveness of a home hazard modification program for reducing falls in urban community-dwelling older adults: A randomized controlled trial. Japan Journal of Nursing Science. 2015;12(3):184-97.
〇 34) Kamei T, Kajii F, Yamamoto Y, Irie Y, Kozakai R, Sugimoto T, et al. Effectiveness of a home hazard modification program for reducing falls in urban community-dwelling older adults: A randomized controlled trial. Japan Journal of Nursing Science. 2015; 12(3):184-97.

Citation:
× Tomoko et al.(2015)
〇 Kamei et al.(2015)

Furthermore, there are errors in the numerical values listed in the tables3. In addition, the flow of literature selection is not easy to understand noted in Figure 1.
Table3
Tomoko K et al.(2015) Intervention Group
× 56F 11M
〇 54F 9M

---

## Round 0.2 · accepted · Accept

Please change "the older adult" to "older adults in paragraph 1 of the Introduction. Lines 46 and 51 on the tracked changes version.

Reviewer 1 ·

Basic reporting

The authors have made several recommended revisions that have strengthened the manuscript.

Experimental design

no comment

Validity of the findings

no comment

Additional comments

I recommend publishing this important study. Thank you for allowing me the privilege of reviewing.